# Effect of Curcumin as Feed Supplement on Immune Response and Pathological Changes of Broilers Exposed to Aflatoxin B1

**DOI:** 10.3390/biom12091188

**Published:** 2022-08-26

**Authors:** Sihong Li, Meiyu Han, Yixin Zhang, Muhammad Ishfaq, Ruimeng Liu, Gaoqiang Wei, Xiuying Zhang, Xiuying Zhang

**Affiliations:** 1Heilongjiang Key Laboratory for Animal Disease Control and Pharmaceutical Development, Faculty of Basic Veterinary Science, College of Veterinary Medicine, Northeast Agricultural University, 600 Changjiang Road, Heilongjiang 150030, China; 2Key Laboratory of Applied Technology on Green-Eco-Healthy Animal Husbandry of Zhejiang Province, College of Animal Science and Technology, College of Veterinary Medicine, Zhejiang A&F University, Hangzhou 311300, China; 3China Institute of Veterinary Drug Control, Beijing 100081, China

**Keywords:** curcumin, aflatoxin B1, broiler, immune response, pathological changes

## Abstract

In this study, we examined the protective effects of curcumin against the AFB1-induced immune response of and pathological changes in broilers. Histopathology examinations showed that at day 28, AFB1 (5 mg/kg) exposure leads to severe histological changes in the spleen, thymus and bursa of Fabricius with a decrease in the number and karyoplasmic area ratio of plasma cells. Curcumin alleviated the AFB1-induced immune organs’ damage as well as the changes in plasma cells in a dose-dependent manner. RT-PCR data showed that AFB1 significantly downregulated the IL-2 and IFN-γ mRNA expression levels in the thymus, spleen and bursa of Fabricius. However, curcumin supplementation improved the AFB1-induced immune organs’ damage via upregulated cytokines’ expression. Intriguingly, similar trends were noticed in abnormal morphological changes and the immune response at day 35 after the withdrawal of AFB1 and curcumin from the diet, suggesting the protective effects and immunomodulatory function against AFB1 in broilers. The current study provides a scientific experimental basis for the application of curcumin as a therapeutic drug or additive in animal husbandry productive practice.

## 1. Introduction

Aflatoxicosis is a kind of serious animal disease that has become a global problem threatening food safety and livestock health. The sudden deaths of 100,000 turkeys in the United Kingdom in the 1960s were linked to peanut meal imported from Brazil [1]. Further investigation showed that the peanut meal and cereals were contaminated with a toxic metabolite from Aspergillus flavus [2,3]. Aflatoxin B1 (AFB1) is the most toxic metabolite that was classified by the International Agency for Research on Cancer (IARC) as a group I carcinogen in 1993 [4]. AFB1 can cause acute and chronic poisoning, as well as cancer, in numerous animals and humans due to its carcinogenic, teratogenic and mutagenic effects [5,6]. The intake of AFB1 lowers the resistance of animals to bacteria, viruses, parasites and other diseases, and increases the susceptibility to diseases. In commercial poultry species, an AFB1-contaminated diet results in poor performance, high mortality, liver damage and, most notably, immunosuppression [7].

In poultry, central and peripheral tissues such as the thymus, spleen and bursa of Fabricius play critical roles in the body’s defense against invading pathogens [8]. The spleen is the largest immunological organ in the body [9]. T lymphocytes develop and mature in the thymus, which is part of the central immune system, while B lymphocytes develop and mature in the bursa of Fabricius (which is specific to birds) [10]. The excessive intake of AFB1 may disrupt immune organ growth and development, resulting in a reduction in the size of immune organs. In serious cases, immune organs might atrophy, endangering the health and productivity of poultry [11]. AFB1 can cause cellular and humoral immunosuppression by the reduction in the number of white blood cells and the level of immunoglobulin [12,13]. AFB1 also alters immunity by increasing lymphocytes’ apoptosis [14], inhibiting lymphocytes’ activation [15], lowering lymphocytes in the peripheral circulation and downregulating cytokines [16]. Therefore, it is critical to completely comprehend the damage and toxicity caused by AFB1 to the broiler immune system, as well as to develop efficient countermeasures to AFB1-induced immunotoxicity.

Curcumin belongs to a kind of phenolic pigment, which is the main active component of turmeric [17]. Many studies have shown that curcumin displays anti-inflammatory, antitumor, antioxidant, free radical scavenging and immune system enhancement capacities [18]. The dietary supplementation of curcumin markedly improved the production performance caused by AFB1 in broilers and the body weight gain of broilers was significantly increased by the oral administration of 150 mg/kg, 222 mg/kg and 444 mg/kg of curcumin [19,20]. Our team’s previous researches have proved that curcumin can be used as a hepatic protectant to reduce oxidative stress, enhance antioxidant capacity, regulate metabolism and for the detoxification of AFB1 to attenuate AFB1-induced toxicity in the liver [21,22]. It has been proved that curcumin can restore AFB1-induced alterations in immunoglobulin levels and cytokines’ gene expression in duckling spleens, therefore enhancing their immunological function [23]. Furthermore, a diet supplemented with curcumin promoted B and T lymphocyte proliferation in the spleen, suggesting that curcumin has a positive effect on the immunomodulation of broiler chickens [24]. However, there is little research on curcumin that shows it could alleviate the immune response and pathological changes induced by AFB1 in various immune organs comprehensively. To evaluate the protective role of curcumin on the AFB1-induced immune response of and histological changes in broilers’ whole immune system, the histological observation, viscera index, number and karyoplasmic area ratio of plasma cells, as well as cytokines’ gene expression, were determined in broilers’ thymus, spleen and bursa of Fabricius. The current study will enrich the research data of aflatoxicosis in the immune system of poultry and provide a scientific experimental basis for the application of curcumin as a therapeutic drug or additive in animal husbandry productive practice.

## 2. Materials and Methods

### 2.1. Chemicals and Reagents

Curcumin powder (2.5%) was bought from Henan ShengXing Biological Company Co., Ltd. (Henan, China). Standard AFB1 (purity ≥ 98%) was purchased from Sigma-Aldrich Co., Ltd. (Shanghai, China). Formaldehyde, ethanol, dimethylbenzene, acetone, paraffin, hematoxylin, eosin and neutral balsam were supplied from Tianjin Yongda Chemical Reagent Company Co., Ltd. (Tianjin, China). Methyl green and pyronin were purchased from Biosharp Biological Company Co., Ltd. (Hefei, China).

### 2.2. Chickens and Treatment

In this experiment, ninety-six commercial unsexed Arbor Acres (AA) broilers (one-day-old) were obtained from Yi Nong Commercial hatchery (Heilongjiang, China, registration number: 230108799294096). All animal procedures were performed in compliance with the Guidelines for Care and Use of Laboratory Animals, 8th edition [25], and approved by the Animal Ethics Committee of Northeast Agricultural University, China. All broilers were provided free access to feed, fresh clean water and 12:12 h light/dark cycle. After 3 days of acclimatization, the broilers were separated into six groups (*n* = 16) randomly and provided a diet for the next 28 days as follows: Control (blank control diet); Curcumin control (450 mg/kg curcumin); AFB1 (5 mg/kg AFB1); Curcumin I (150 mg/kg curcumin + 5 mg/kg AFB1); Curcumin II (300 mg/kg curcumin + 5 mg/kg AFB1); Curcumin III (450 mg/kg curcumin + 5 mg/kg AFB1), while from day 28 to day 35, blank control diet was provided for all six groups. AFB1 was dissolved in 30 mL methanol, then this prepared solution was sprayed evenly on the basal eat feed and mixed to obtain the 5 mg/kg AFB1-contaminated diet [26]. The equivalent methanol was sprayed evenly on the normal feed to obtain the basal diet. The treatment concentration of curcumin was calculated and added uniformly into the diet and mixed evenly. For instance, 18 g curcumin (2.5%) powder was weighed and added into the basal diet, then the diet was mixed thoroughly to obtain the 450 mg/kg curcumin-contaminated diet. All the diets were evaporated at 37 °C and mixed with a vertical mixer.

### 2.3. Sample Collection and Viscera Index

At day 28 and 35, eight broilers in each group were euthanized by carbon dioxide (CO2) asphyxiation, respectively. Thymus, spleen and bursa of Fabricius tissues were excised and divided into two parts: a part of tissue was washed with phosphate buffer solution (PBS, pH = 7.4), prepared as 1 cm × 1 cm and fixed with 4% formaldehyde for histological examination, and the remaining was immediately frozen in liquid nitrogen and stored at −80 °C for further analysis. Clinical observations were performed throughout the experimental period. The body weights (kilogram) and viscera weights (gram) were recorded. The viscera index (VI) of thymus, spleen and bursa of Fabricius was computed using the formula: VI = Viscera weight/Body weight × 100%.

### 2.4. Histological Observation

The thymus, spleen and bursa of Fabricius were processed for hematoxylin and eosin (H.E.) staining. The tissues were fixed in 4% formaldehyde for 24 h at 4 °C and then washed overnight with current water. They had dehydration treatment with graded ethanol vitrification with dimethylbenzene and were deposited in paraffin. The thymus, spleen and bursa of Fabricius were cut into 4 μm slices by rotary microtome (Leica, Germany), and then stained with hematoxylin–eosin. The tissue slices were sealed with neutral balsam mounting medium (Tianjin Yongda, China) and examined using micropathological imaging system (Nikon E100, Tokyo, Japan).

### 2.5. Counting and Karyoplasmic Area Ratio of Plasma Cells

The obtained tissue slices (4 μm) of thymus, spleen and bursa of Fabricius were stained with methyl green-pyronin. The sections were rinsed with distilled water after staining, dehydrated orderly with anhydrous acetone, xylene–acetone mixture and pure xylene, finally sealed with neutral balsam mounting medium. The number of plasma cells in each lymphoid organ was examined with a Nikon microscope under 100× magnification using immersion oil. For plasma cells’ counting, seven fields from each tissue slice were chosen randomly. For the analysis of karyoplasmic area ratio, at least 15 plasma cells were randomly selected from each immune organ tissue section.

### 2.6. RNA Extraction and cDNA Synthesis

Briefly, total RNA was extracted by using TRIZOL (Sigma-Aldrich, Shanghai, China) according to the instructions. A260/A280 ratio was detected using Nanodrop 2000c spectrophotometer (Thermo Scientific, Waltham, MA, USA) to evaluate RNA quality and concentration [27]. The first strand cDNA was synthesized by using ReverTra Ace qPCR RT Master Mix (Toyobo Co Ltd., Osaka, Japan).

### 2.7. Quantitative Real-Time PCR (qPCR) Analysis

Quantitative Real-time PCR (qPCR) was conducted according to the previous procedure used by Wang et al. [28]. Specific gene primers (Table 1) for interleukin-2 (IL-2) and interferon-gamma (IFN-γ) were designed by Primer 5.0 based on the gene sequence in NCBI database. GAPDH was used as housekeeping gene as control for normalization. The qPCR was performed for determination of IL-2 and IFN-γ mRNA expression by using FastStart Universal SYBR Green Master ROX (Roche, Shanghai, China). The amplification program was as follows: pre-denaturation at 95 °C for 10 min, and then 40 cycles of denaturation at 95 °C for 15 s and annealing at 58 °C for 30 s, followed by extension at 95 °C for 15 s, finally extension at 37 °C for 30 s and cooling at 4 °C. The mRNA expression levels were calculated by the 2−ΔΔCT method [29].

### 2.8. Statistical Analysis

All experiments were repeated in triplicate to ensure accuracy and reproducibility. All the data were expressed as means ± standard deviation (SD) and analyzed by SPSS (version 19.0, Chicago, IL, USA). The significant difference among the experimental groups was determined by using one-way analysis of variance. Values of *p* < 0.05 were considered as statistically significant. The figures were made by GraphPad prism (version 5.0, San Diego, CA, USA).

## 3. Results

### 3.1. Clinical Observation of Arbor Acres Broilers’ Livers

There were no deaths of broilers during this experiment, but all broilers in the AFB1 group were significantly smaller in body size with a decreased appetite, delayed growth, depressed behavior, dull eyes, and loose and dull feathers. No chicken died and no symptoms were observed in the control group and curcumin control group. The clinical signs of the AFB1 group disappeared in the curcumin groups I, II and III with good growth behavior (Table 2).

### 3.2. Effects on Viscera Index of Thymus, Spleen and Bursa of Fabricius

At day 28, there were significant reductions in the viscera index of the thymus, spleen and bursa of Fabricius in the AFB1 group compared to the control (Figure 1). Additionally, it is noted that the AFB1-induced viscera index reduction could not be self-recovered at day 35. In the curcumin groups II and III, the viscera indexes were obviously ameliorated, and the ameliorative effect of curcumin was continuous up to 7 days after the withdrawal of curcumin from the diet. In addition, the ameliorative effect of curcumin on promoting the growth and development of the immune organs via an increasing viscera index in broilers was in a dose-dependent manner.

### 3.3. Histopathological Changes in Immune Organs

#### 3.3.1. Histopathological Changes in Thymus

On days 28 and 35, the reticular cells in the broilers’ thymic cortical area from the control group and curcumin control group showed a clear morphology with a small number of nuclear debris around (Appendix A). At day 28, the broilers fed with AFB1 showed an abnormal morphology of the reticular cells in the thymus’ cortical area surrounding nuclear fragments. Meanwhile, obvious congestion and cavities were observed in the medullary area (Appendix A), whereas all curcumin dose groups showed different degrees of remission in the thymic cortex (Appendix A). In the curcumin III group, thymus pathological injury was partially alleviated (Appendix A). At day 35, a histopathological lesion in the thymus was not self-recovered in the AFB1 group (Appendix A). It is clear that after the withdrawal of AFB1 and curcumin from the diet, the therapeutic effect of curcumin still remains for up to 7 days in a dose-dependent manner (Appendix A).

#### 3.3.2. Histopathological Changes in Spleen

On day 28 and 35, broilers’ spleen cells from the control group and curcumin control group were arranged in order and revealed no abnormal changes (Appendix A). At 28 days, there was the appearance of severe congestion in the spleen red pulp area and a reduction in the lymphocytes in the periarteriolar lymphoid sheath in the spleen from broilers fed with AFB1. Compared to day 28, the spleen lesions of the AFB1 group were not improved at day 35 (Appendix A). Low and medium doses of curcumin show interventional effects on congestion and the lymphocytes’ decrease in the spleen (Appendix A), and a high dose (450 mg/kg) of curcumin was the most effective (Appendix A). Furthermore, the effect of curcumin against AFB1 continued for 7 days after its withdrawal from the diet in a dose-dependent manner (Appendix A).

#### 3.3.3. Histopathological Changes in Bursa of Fabricius

On days 28 and 35, the cortex and medulla cells were neatly arranged with a clear lymphocytes’ cortico-medullary border in the broilers’ bursa of Fabricius from the control group and curcumin control group (Appendix A). The bursa of Fabricius revealed a sparseness and disarrangement of the lymphocytes’ medullary area in the AFB1 group as well as a large number of small circular voids in which numerous nuclear fragments were found (Appendix A). Compared to 28 days, the bursa of Fabricius in the AFB1 group at 35 days showed more severe lesions and nuclear fragments (Appendix A). Curcumin alleviated AFB1-induced bursa of Fabricius damage in a dose-dependent manner. Moreover, the preventive effect of curcumin was shown after 7 days of its withdrawal (Appendix A).

### 3.4. Effects on Number and Karyoplasmic Area Ratio of Plasma Cells in Immune Organs

The number of plasma cells in the body can reflect humoral immunity to some extent. On days 28 and 35, there was no evidence of changes in the number of plasma cells in the thymus, spleen and bursa between the control group and curcumin control group. In the AFB1 groups, a significant reduction was observed in the plasma cells’ numbers in the thymus, spleen and bursa of broilers relative to the control group (Figure 2). The amount of plasma cells in the thymus and bursa of broilers in the curcumin group I on day 28 did not differ significantly from the AFB1 group; however, there was a considerable increase in the spleen. Notably, at day 28, dietary curcumin at 300 mg/kg and 450 mg/kg significantly increased the number of plasma cells in the thymus, spleen and bursa of broilers (*p* < 0.05). Furthermore, the anti-AFB1 effects of curcumin lasted for 7 days after the withdrawal of curcumin from the diet. With the rise in dose, the ameliorative impact of curcumin on raising the number of plasma cells in the thymus, spleen and bursa become more noticeable (Figure 2).

The karyoplasmic area ratio of plasma cells can indirectly reflect the proliferation capacity of plasma cells, which is widely used in the classification and differential diagnosis of tumors. The karyoplasmic area ratio of normal plasma cells is maintained within a certain range; when plasma cells are damaged, the value of the ratio shows abnormal changes. Compared to the control group, the present results (Figure 3) show that the karyoplasmic area ratios of the plasma cells in the thymus, spleen and bursa of broilers treated with AFB1 were significantly decreased (*p* < 0.05) at day 28. The karyoplasmic area ratio of the plasma cells in immunological organs cannot be recovered by removing AFB1 from the diet. Low dose curcumin (150 mg/kg) was not sufficient to alleviate the AFB1-induced reduction in the karyoplasmic area ratio in the thymus (*p* > 0.05, Figure 3A). In immune organs, both a medium and high dose of curcumin successfully caused a significant recovery of the karyoplasmic area ratio in a dose-dependent manner (*p* < 0.05). Meanwhile, the same trend of curcumin ameliorating the plasma cell karyoplasmic area ratio was found at day 35 after the withdrawal of AFB1 and curcumin from the diet.

### 3.5. Effect of AFB1 and Curcumin on Cytokines’ Expression

On days 28 and 35, the IL-2 and IFN-γ mRNA expressions were measured in broilers to observe the effects of AFB1 and curcumin on cellular immunity (Figure 4 and Figure 5). It was found that the IL-2 and IFN-γ mRNA expressions in the thymus, spleen and bursa were decreased in the AFB1 group (*p* < 0.05) at day 28. However, the withdrawal of AFB1 did not improve the inhibition of IL-2 and IFN-γ mRNA levels at day 35. In the curcumin I group (150 mg/kg curcumin) the expressions of IL-2 or IFN-γ mRNA in the thymus and spleen were not significantly improved Figure 4A,B and Figure 5A,B) but significantly increased in the bursa of Fabricius (*p* < 0.05) compared with the AFB1 group (Figure 4C and Figure 5C). The 300 mg/kg and 450 mg/kg curcumin supplementation obviously (*p* < 0.05) prevents immune organs from developing AFB1-induced injury by upregulating the mRNA expressions of IL-2 and IFN-γ in a dose-dependent manner at 28 and 35 days.

## 4. Discussion

Aflatoxin B1 is the most common aflatoxin contamination, which seriously affects the safety of animal food and human health through the food chain [30,31]. Curcumin is a polyphenolic compound extracted from turmeric and has a wide range of pharmacological actions, including antiinflammatory, antioxidant and antitumor properties [18,32]. Curcumin has attracted great interest as a green and novel feed additive in recent years. However, the immune response induced by AFB1 lacks comprehensive study. We studied the impact of AFB1 on pathological alterations and the immunological response in the thymus, spleen, and bursa of Fabricius, as well as the ameliorative effects of curcumin, to better understand the protective effect of curcumin against AFB1-induced immune system harm. The central and peripheral lymphoid tissues of birds play an important role in the body’s defense against pathogens [8]. The thymus is the primary organ where T lymphocytes develop and mature, and it is a part of the central immune system. The function of the thymus is to induce the differentiation of immature T cell precursors from the bone marrow, spleen and other lymphoid tissues into immune-active T lymphocytes, which can generate a cellular immune response and play an immune role after being stimulated by an antigen [33]. The spleen is the secondary lymphoid organ, which contains the red and white pulp. The red pulp produces and stores red blood cells, and the white pulp plays a role in the immune system [34]. The bursa of Fabricius is a unique structure of birds, where B lymphocytes develop and mature to produce antibodies to participate in the body’s immune response [10].

We found that at days 28 and 35, AFB1 exposure resulted in a decrease in the viscera index of the thymus, spleen and bursa of Fabricius and an impairment of the immune organs’ development. There is evidence that AFB1 has a certain inhibitory effect on the growth and development of immune organs in broiler chickens [35,36]. In other studies, where diets contained lower amounts of AFB1 and OTA, body weight differences were observed due to the low doses of mycotoxins used alone or in combination [37,38]. Giambrone found that 200 ppb of AFB1 resulted in a significant decrease in turkeys’ cell-mediated immunity [39], while 1000 ppb of pure AFB1 given to 2-week-old broilers for 5 weeks caused a mildly toxic effect [40]. Importantly, in our pre-experiments, we administered 1, 3 and 5 mg of AFB1 per kg of feed. On the basis of our preliminary data and the previous literature [41], we choose a dose of 5 mg/kg of AFB1. The immune organ viscera index of the curcumin I, II and III groups increased in a dose-dependent manner at days 28 and 35, indicating that curcumin could promote the growth and development of the immune organs in broilers, which corroborates the findings of other studies [23,42]. Notably, after AFB1 and curcumin withdrawal, the inhibitory effects of AFB1 and the ameliorative effects of curcumin on the immune organs’ viscera index still existed at day 35.

In this study, the previously reported injurious effects of AFB1 were reproduced in broilers by feeding them 5 mg/kg of AFB1 in their diet. Histological examination of the lymphoid organs (thymus and bursa of Fabricius) revealed morphological abnormalities in the cortical and medullary areas with nuclear fragments and cavities, severe hyperemia in the red medulla area of the spleen, as well as significantly smaller lymphatic follicles. Expectedly, the pathological changes caused by AFB1 still remained, and were even aggravated in some broilers at day 35. This indicates that the immune organ injury caused by AFB1 cannot be self-recovered even if the withdrawal of the administration of AFB1 and the immunotoxicity of AFB1 has a post effect. Previously, the immunotoxicity effects of dietary AFB1 have been reported in poultry. After 6-day-old Turkeys were fed with AFB1 and AFB2 for 2 weeks, microscopy examination showed that a severe depletion of the lymphoid cells occurred in the bursa of Fabricius and spleen [43]. The broilers’ diet with 0.3 mg/kg of AFB1 for 2 weeks induced histopathological lesions and a lower relative weight in the bursa of Fabricius [44]. However, there were also contrary reports. According to Ortatatli [45], there was no statistical difference in the relative weight of the spleen between aflatoxins-exposed and control broiler chicks. This disparity might be related to the different types of toxins. Thus, our research demonstrated that the decrease in lymphocytes and tissue atrophy induced by AFB1 can lead to a decrease in the immune organ viscera index, indicating the sensitivity of immune organs to AFB1. It has been reported that a diet with 0.15% yeast cell walls led to a decrease in the severity of the liver’s and immune organ’s histological lesions caused by mycotoxins (AFB1, OTA, AFB1 + OTA) [46]. An amount of 0.6 mg/kg selenium (Se) effectively protected broilers from AFB1-induced thymus pathological lesions [14]. We found that supplementary curcumin alleviates AFB1-induced immune organs’ injury in a dose-dependent manner at both day 28 and 35, suggesting that curcumin has aftereffects and beneficial effects on the development of the immune organs and the immune function.

Plasma cells produce antibodies after being triggered by antigens or cytokines in order to boost the body’s humoral immunity. A plasma cell’s karyoplasmic area ratio was also the first feature of relevance in the automatic classification of abnormal lymphoid cells [47]. A normal plasma cell’s karyoplasmic area ratio is maintained within a certain range. When cells are damaged, such as in tumor disease [48], the nuclear–plasma area ratio varies abnormally. Therefore, the number and karyoplasmic area ratios of plasma cells, to some extent, can reflect the humoral immunity. Zhao found that dietary AFB1 supplementation induced spleen injury, as seen by a reduced number of lymphocytes, a smaller region of white pulp and reduced histiocyte proliferation [49]. As expected, AFB1 exposure remarkably reduced the number and karyoplasmic area ratios of plasma cells, which implies that AFB1 can cause the body severe humoral immune suppression by destroying a large number of B lymphocytes to cause apoptosis or necrosis and affect the generation and development of plasma cells. Furthermore, curcumin increases the number and the karyoplasmic area ratios of plasma cells in a dose-dependent manner and a diet of 450 mg/kg of curcumin has the best remission effect against AFB1. It may be that curcumin can stimulate the B lymphocytic proliferation and differentiation [50] to produce plasma cells, and then plasma cells synthesize and secrete antibodies. This significantly increases of the intracytoplasmic organelles’ content and the enhancement of the activity of major organelles related to protein synthesis including the Golgi apparatus, endoplasmic reticulum and ribosomes. The cytoplasm expands continuously due to the increase in various components resulting in the karyoplasmic area ratio of plasma cells decreasing gradually. In general, curcumin can promote the growth and development of plasma cells in immune organs to resist AFB1-induced immune system injury to some extent.

Cytokines, such as IL-2 and IFN-, are polypeptide factors secreted and released by T lymphocytes (CD4+ or CD8+), which can regulate cell activity and play a critical role in cellular immunity [51,52,53]. IL-2 not only promotes B lymphocyte proliferation and differentiation, as well as antibody secretion, but it also enhances antiviral activity by suppressing CD8 + T cells’ apoptosis [54]. IFN-γ, a soluble glycoprotein with high efficiency and activity, is a cytokine with antitumor, antivirus and strong immunomodulatory capabilities that can directly induce T and B lymphocytes’ development [55,56]. We observed that AFB1 exposure considerably downregulated the expressions of IL-2 and IFN-γ mRNA. This is in accordance with previous reports [57,58] in that the expression levels of the IL-2 and IFN-γ genes are associated with AFB1-induced immune disorder, probably due to AFB1 poisoning causing DNA transcription and mRNA translation in lymphocytes and ultimately affecting protein biosynthesis [59]. In our study, curcumin recovered the mRNA expressions of IL-2 and IFN-γ in a dose-dependent manner, and a diet of 450 mg/kg of curcumin had the best remission effect against AFB1. These results indicated that curcumin could activate the cytokine genes’ expression in immune organs against AFB1-induced immune system injury. Furthermore, we discovered that the immunotoxicity of AFB1 and the protective effects of curcumin lasted until day 35, implying that AFB1 and curcumin have aftereffects and beneficial effects on broiler growth and immune function.

## 5. Conclusions

This research for the first time gives a comprehensive insight into the immunotoxicity of AFB1 to the thymus, spleen and bursa of Fabricius in broilers, as well as the ameliorative effects of curcumin. Curcumin diets successfully protected the immunological organs from AFB1-induced tissue damage and immunosuppression by lowering plasma cell numbers and the karyoplasmic area ratio with the downregulation of cytokines. Additionally, the ameliorative effects of curcumin were in a dose-dependent manner and had an aftereffect. The current study is about AFB1-induced immunological damage, and curcumin can be utilized as a feed additive to protect broiler immune systems from AFB1 toxicity.

## Figures and Tables

**Figure 1 biomolecules-12-01188-f001:**
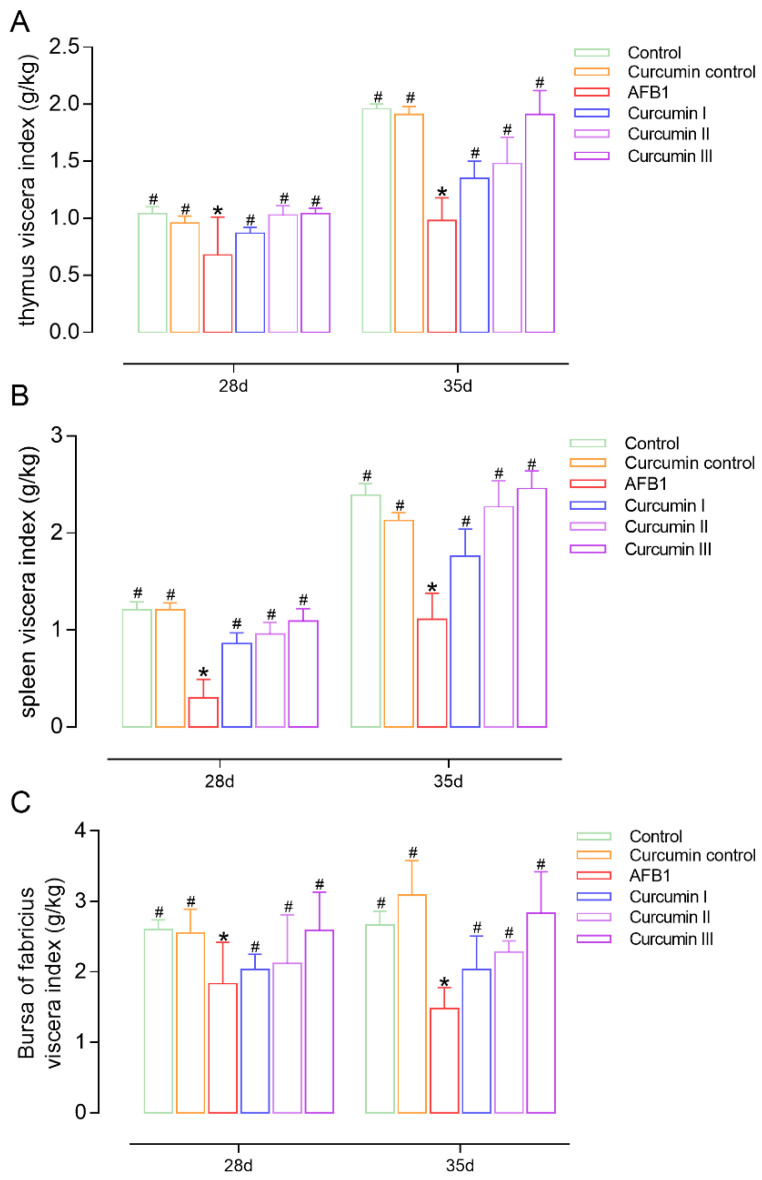
Change in viscera index (g/kg) of broiler chickens at 28 days and 35 days (*n* = 8). (**A**) Thymus viscera index, (**B**) spleen viscera index, (**C**) bursa of Fabricius viscera index. Each bar represents means ± standard deviation (SD). * *p* < 0.05 shows the statistically significant differences from control group. ^#^
*p* < 0.05 shows the statistically significant differences from AFB1 group.

**Figure 2 biomolecules-12-01188-f002:**
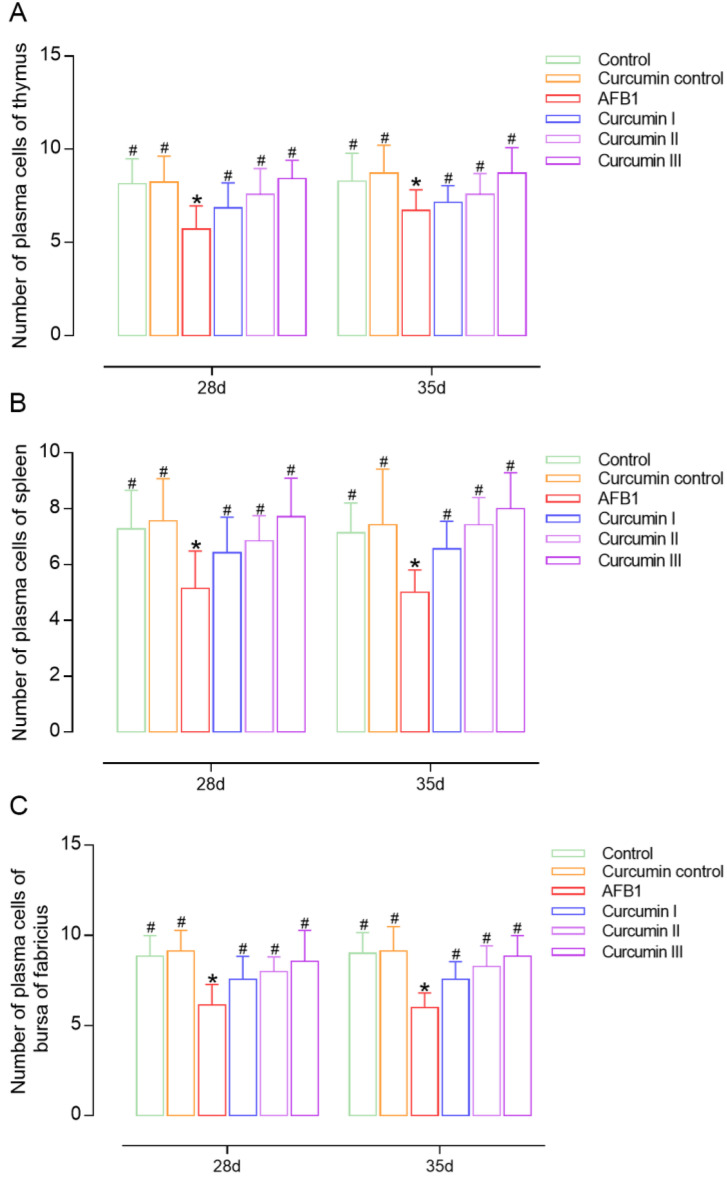
Change in plasma cell numbers in the immune organs of broiler chickens at 28 days and 35 days (*n* = 8). (**A**) Thymus plasma cell number, (**B**) spleen plasma cell number, (**C**) bursa of Fabricius plasma cell number. Each bar represents means ± standard deviation (SD). * *p* < 0.05 shows the statistically significant differences from control group. ^#^
*p* < 0.05 shows the statistically significant differences from AFB1 group.

**Figure 3 biomolecules-12-01188-f003:**
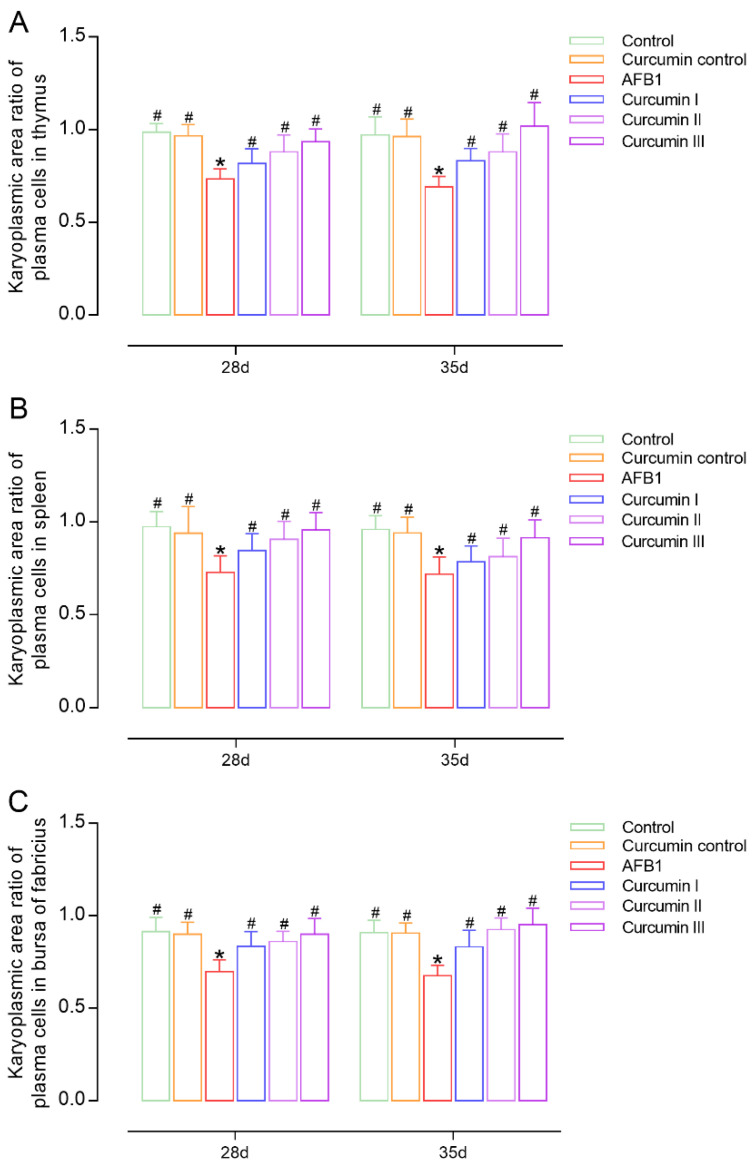
Change in karyoplasmic area ratio of plasma cells in the immune organs of broiler chickens at 28 days and 35 days (*n* = 8). (**A**) Thymus karyoplasmic area ratios of plasma cells, (**B**) spleen karyoplasmic area ratios of plasma cells, (**C**) bursa of Fabricius karyoplasmic area ratios of plasma cells. Each bar represents means ± standard deviation (SD). * *p* < 0.05 shows the statistically significant differences from control group. ^#^
*p* < 0.05 shows the statistically significant differences from AFB1 group.

**Figure 4 biomolecules-12-01188-f004:**
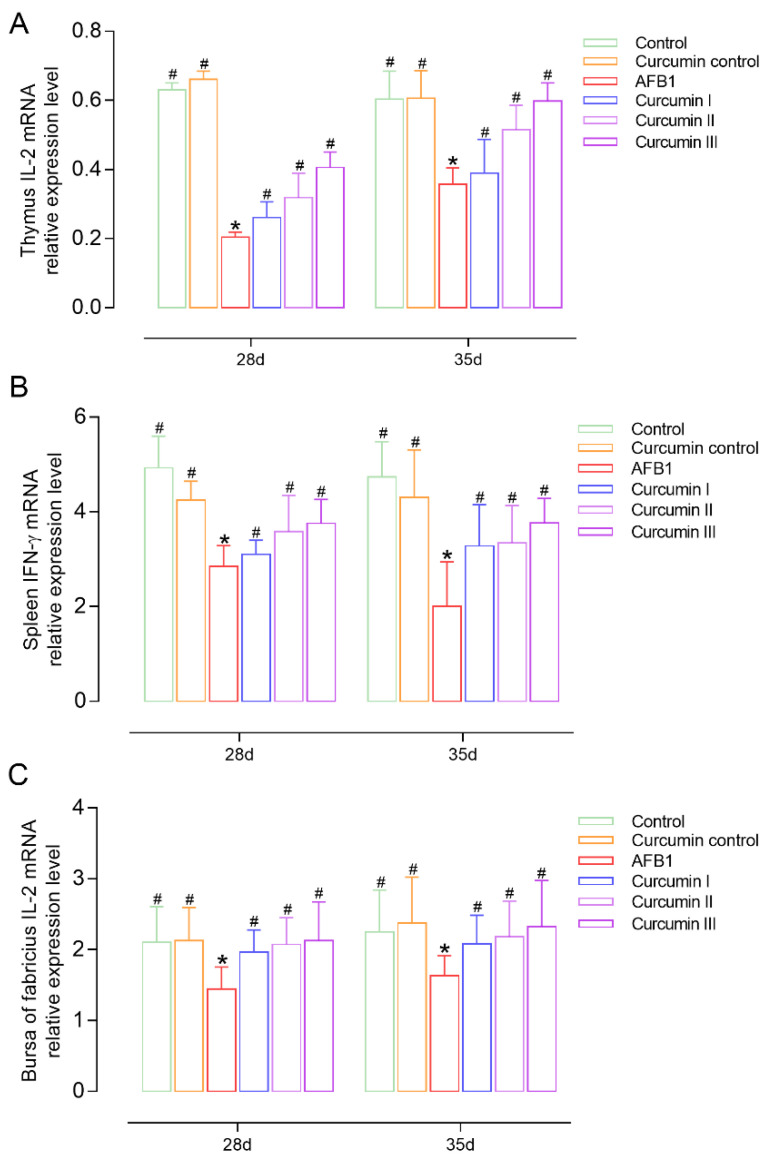
Changes in the expression of IL-2 in the immune organs of broiler chickens at 28 days and 35 days (*n* = 8). (**A**) The IL-2 mRNA expression in thymus, (**B**) the IL-2 mRNA expression in spleen, (**C**) the IL-2 mRNA expression in bursa of Fabricius. Each bar represents means ± standard deviation (SD). * *p* < 0.05 shows the statistically significant differences from control group. ^#^
*p* < 0.05 shows the statistically significant differences from AFB1 group.

**Figure 5 biomolecules-12-01188-f005:**
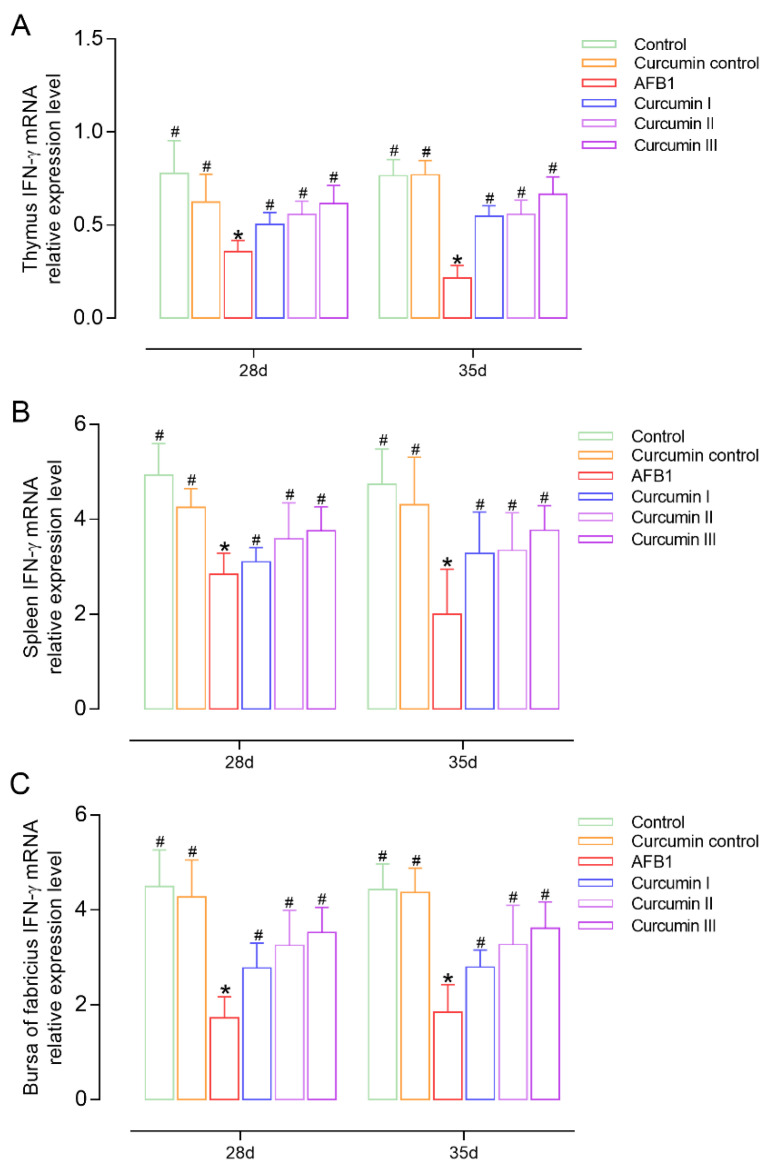
Changes in the expression of IFN-γ in the immune organs of broiler chickens at 28 days and 35 days (*n* = 8). (**A**) The IFN-γ mRNA expression in thymus, (**B**) the IFN-γ mRNA expression in spleen, (**C**) the IFN-γ mRNA expression in bursa of Fabricius. Each bar represents means ± standard deviation (SD). * *p* < 0.05 shows the statistically significant differences from control group. ^#^
*p* < 0.05 shows the statistically significant differences from AFB1 group.

**Table 1 biomolecules-12-01188-t001:** Specific gene primers used for qPCR.

Gene	Primers (from 5′ to 3′)	Length (bp)
GAPDH	Forward GCACGCCATCACTATCTT	82
	Reverse GGACTCCACAACATACTCAG	
IL-2	Forward GGATCCATGATGTGCAAAGTACTG	80
	Reverse CGGTCGACTTATTTTTGCAGATATCT	
IFN-γ	Forward TCCTTCTGAAAGCTCTCGCC	175
	Reverse CTGGTGTCCAGGATGGTGTC	

**Table 2 biomolecules-12-01188-t002:** Clinical observation of Arbor Acres broilers.

Group	Body Size	Eyes	Feathers	Behavior
Control	normal	bright eyes	glossy feathers	normal appetite, growth and behavior
Curcumin Control	normal or bigger	bright eyes	glossy feathers	normal appetite, growth and behavior
AFB1	smaller	dull eyes	dull feathers	decreased appetite, delayed growth and depressed behavior
Curcumin I	normal	bright eyes	glossy feathers	normal appetite, growth and behavior
Curcumin II	normal	bright eyes	glossy feathers	normal appetite, growth and behavior
Curcumin III	normal	bright eyes	glossy feathers	normal appetite, growth and behavior

## Data Availability

Not applicable.

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
