# Peer review of "Effect of Curcumin as Feed Supplement on Immune Response and Pathological Changes of Broilers Exposed to Aflatoxin B1"

_biomolecules, 2022, doi:10.3390/biom12091188_

Round 1

Reviewer 1 Report

This manuscript entitled “Effect of Curcumin as Feed Supplement on Immune Response and Pathological Changes of Broilers Exposed to Aflatoxin B1” suggests that curcumin has benefitial effects against AFB1-mediated immunosuppression. However, results are poorly presented and there are several major concerns throughout the text.

Major concerns:

- The experimental design is not correct. The control groups with 150 and 300 mg/kg of curcumin were not included. The authors should justify the chosen design.

- There are serious errors in the description of the methods.

- The figures are poor and must be improved. Its quality is very bad and the legends are incomplete and incorrect. It is very difficult the interpretation of the graphs. Include colors or different patterns.

- The results are not clear described. It is difficult for the reader to be able to follow the ideas raised by the authors. 

Comments to the authors:

- It is not clear in the abstract the aim and the hypothesis of this work.

-Line 37. Typo “atatoxin”

-Regarding AFB1 treatment in materials and methods section the authors state that “…prepared solution was sprayed evenly on the basal feed and mixed to obtain the 5 mg/kg AFB1-contaminated diet”. How did the authors ensure that they standardize treatment? How did they control that the spray was homogeneous and that the correct concentration was reached?

-Regarding curcumin treatment: What method was used to incorporate curcumin into the basal feed? Please, add this information in 2.2 section.

-Authors must specify the exact time of fixation. In 2.4 section is stated that “The tissues were fixed in 4% formaldehyde for more than 24 h at 4℃

-“RNA quality was assessed to detect A260 / A280 ratio using Nanodrop 2000c spectrophotometer”. Please, improve this sentence. The aim is not to detect A260/A280, is to evaluate RNA quality and concentration.

-“The first strand cDNA was reverse transcribed by using ReverTra Ace qPCR RT Master Mix” The RNA was reverse transcribed, not the cDNA.

-Please, add a table with the data of body size of each group as well as appropriate observations about eyes, feathers, growth behavior, etc.

-Figure 1 (2 and 3). The legend is incomplete and has errors: Authors used 16 broilers in total, but in each group n=8. Concerning statistical analyses, it refers to “different letters” however there are not letters.

-Supp Figures. There are no bars however in the legend it says “(Bar = 50 μm)

-Figure 2. All the bars present a lot of deviation and do not appear to be significantly different however there are asterisks and hashtags. Please, add in the legend the statistical analysis performed in each experiment.

In summary, considering the above mentioned major concerns, I believe this present form of the manuscript is not suitable for publication in Biomolecules.

Author Response

Point 1: The experimental design is not correct. The control groups with 150 and 300 mg/kg of curcumin were not included. The authors should justify the chosen design.

Response 1: Thanks for your nice comments. 450 mg/kg Curcumin was used in curcumin control group because this was the highest intervention concentration used in this experiment. The broilers in the curcumin control group (450 mg/kg) showed good growth behavior and healthy, indicating that curcumin at concentrations of 150 mg/kg and 300 mg/kg was a very safe antidote against AFB1 for broilers without toxicity.

Point 2: There are serious errors in the description of the methods.

Response 2: Thanks for your nice comments. The description of the methods was thoroughly revised and modified accordingly. The errors were corrected (please refer to the part of methods).

Point 3: The figures are poor and must be improved. Its quality is very bad and the legends are incomplete and incorrect. It is very difficult the interpretation of the graphs. Include colors or different patterns.

Response 3: Thanks for your nice comments. We have modified the figures to improve its quality according to the journal guidelines (300 dpi). The legends and colors were carefully checked and corrected. The corresponding notes have been adjusted to match the figures. Please refer to the Figures.

Point 4: The results are not clear described. It is difficult for the reader to be able to follow the ideas raised by the authors.

Response 4: Thanks for your nice comments. We have revised the writing of the results section to make the results better understood by the readers. Additionally, a few description errors were also found and corrected.

Point 5: It is not clear in the abstract the aim and the hypothesis of this work.

Response 5: Thanks for your precious comments on this paper. In the revised paper, we have remodified the abstract part to eliminate unnecessary descriptions and emphasize the aim, hypothesis and significance of research, which can better reflect the value of the work.

Point 6: Line 37. Typo “atatoxin”.

Response 6: Thanks for your nice comments. We have revised the “Atatoxin B1” to “Aflatoxin B1 (AFB1)” (please refer to line 36).

Point 7: Regarding AFB1 treatment in materials and methods section the authors state that “…prepared solution was sprayed evenly on the basal feed and mixed to obtain the 5 mg/kg AFB1-contaminated diet”. How did the authors ensure that they standardize treatment? How did they control that the spray was homogeneous and that the correct concentration was reached?

Response 7: Thanks for your nice comments. First, AFB1 was dissolved in 30 ml methanol, and AFB1 is easily degradable under ultraviolet exposure, so that the prepared solution was sprayed evenly on the basal feed using a sprayer in a dark room and then the diets were mixed with a vertical mixer to obtain the 5 mg/kg AFB1-contaminated diet. The mixed feed is dried to ensure that the methanol is completely evaporated. In the preliminary experiment, the concentration of AFB1 in the feed was determined by HPLC, and the result showed that the concentration of AFB1 could evenly reached to 5.0 mg/kg as shown previously (Cui X, Muhammad I, Li R, et al. Development of a UPLC-FLD Method for Detection of Aflatoxin B1 and M1 in Animal Tissue to Study the Effect of Curcumin on Mycotoxin Clearance Rates. Front Pharmacol. 2017; 8: 650).

Point 8: Regarding curcumin treatment: What method was used to incorporate curcumin into the basal feed? Please, add this information in 2.2 section.

Response 8: Thanks for your nice comments. Curcumin powder can be easily and evenly adsorbed on the feed particles, so just add the calculated curcumin powder and mix the feed thoroughly. For example, 18.0 g curcumin (2.5%) powder was weighed and added into the 1.0 kg basal diet, then the diet was mixed thoroughly to obtain the 450 mg/kg curcumin-contaminated diet. And the method of curcumin-contaminated diet preparation was added in 2.2 section (please refer to line 104).

Point 9: Authors must specify the exact time of fixation. In 2.4 section is stated that “The tissues were fixed in 4% formaldehyde for more than 24 h at 4℃”.

Response 9: Thanks for your nice comments. We have added the exact time in line 124 as“for 24 h at 4℃”(please refer to line 120).

Point 10: “RNA quality was assessed to detect A260/A280 ratio using Nanodrop 2000c spectrophotometer”. Please, improve this sentence. The aim is not to detect A260/A280, is to evaluate RNA quality and concentration.

Response 10: Thanks for your nice comments. We have modified the sentence “RNA quality was assessed to detect A260/A280 ratio using Nanodrop 2000c spectrophotometer” to “A260/A280” ratio was detected using Nanodrop 2000c spectrophotometer (Thermo Scientific, USA) to evaluate RNA quality and concentration” (please refer to line 138).

Point 11: “The first strand cDNA was reverse transcribed by using ReverTra Ace qPCR RT Master Mix” The RNA was reverse transcribed, not the cDNA.

Response 11: Thanks for your nice comments. We have revised the sentence “The first strand cDNA was reverse transcribed by using ReverTra Ace qPCR RT Master Mix” to “The first strand cDNA was synthesized by using ReverTra Ace qPCR RT Master Mix” (please refer to line 139).

Point 12: Please, add a table with the data of body size of each group as well as appropriate observations about eyes, feathers, growth behavior, etc.

Response 12: Thanks for your nice comments. We made a table with the data of body size of each group as well as appropriate observations. Please refer to Table 2 for 3.1.

Point 13: Figure 1 (2 and 3). The legend is incomplete and has errors: Authors used 16 broilers in total, but in each group n=8. Concerning statistical analyses, it refers to “different letters” however there are not letters.

Response 13: Thanks for your nice comments. There were 6 groups in total, and 16 chickens were equally distributed to each group. However, since there were two sample points, 28d and 35d, so 8 chickens were selected for the experiment at 28d and 35d respectively. The legends have been changed from “n=16” to “n=8”, please refer to line 180. And the description of statistical analyses has been revised into “* P < 0.05 shows the statistically significant differences from control group. # P < 0.05 shows the statistically significant differences from AFB1 group” (please refer to line 182). We have checked and corrected the legends of all figures.

Point 14: Supp Figures. There are no bars however in the legend it says “(Bar = 50 μm)”.

Response 14: Thanks for your nice comments. We have checked and modified the legend of all figures. The “(Bar = 50 μM)” has been changed into “(HE, 100X)”.

Point 15: Figure 2. All the bars present a lot of deviation and do not appear to be significantly different however there are asterisks and hashtags. Please, add in the legend the statistical analysis performed in each experiment.

Response 15: Thanks for your nice comments. We have added the description of statistical analysis at the end of the legend (please refer to line 248). And we checked and modified the legend of all figures.

Reviewer 2 Report

* Effect of Curcumin as Feed Supplement on Immune Response 2 and Pathological Changes of Broilers Exposed to Aflatoxin B1*

The article offers relevant information regarding effects of aflatoxin in broilers. The paper needs some improvement in the figures and overall writing. Please see specific comments below.

Abstract

Line 17: I suggest changing “scrutinized” and use “examined” instead.

Line 23: These lines need a correction “histological changes company with”, what does it mean?

Introduction

Line 37:Fix this sentence “Atatoxin B1 (AFB1)”

Line 41: Delete the first period in “efects. [5,6].”

Line 72: Use lower case in “Karyoplasmic area”

**Materials and methods

Line 81: Use “powder” instead of “pulver” in “Curcumin pulver (2.5%) were….”

Line 121: Please describe brand name of “neutral balsam”

  2/.2. Chickens and Treatment/

Line 98-100: Please add the equipment brand name or model used to mix the feed.

Line 108: Change “by” for “with” in “washed by phosphate buffer”.

Line 117: Add the exact time in “for more than 24 h at 4℃”. The statement is vague.

Line 128: “under the oil lens of Nikon microscope” Please rephrase, perhaps to “…organ was examined with a Nikon (model) microscope under 100X magnification using immersion oil”.

Line 133: Must be “Sigma-Aldrich, USA”.

Line 158: Use different wording in “depressed spirit”, maybe lethargy or depressed behavior.

Line 163: Must be “significant reductions” not “were significant decreased”.

Line 186: Must be “ was not _self-recovered_ in AFB1 group”.

Line 207: Eliminate “th” to read “35 days”. Check appropriate use of “th” in all parts of the text.

Line 228: Use lower case T in “group, The present”.

Line 294: “Consist with our results,” needs rephrasing.

Line 322: Add what “Se” stands for, as it is the first time is introduced in the text.

In the discussion there needs to be a brief explanation on why using 5 mg AF/kg feed is relevant. Please add references to support the dosage used and how this can be comparable or not to realistic conditions. Is this frequently observed in some countries? Please add more details.

Figure 1 needs improvement. The asterisks need to be larger and there is a hash (#) instead of different letters denoting significant differences. Same comment for figures 2 to 5.

Please add 1 or 2 relevant histopathological images of lesions.

Relevant pioneering papers need to be included, for instance:

  * Effects of aflatoxin on young turkeys and broiler chickens. J J
    Giambrone, U L Diener, N D Davis, V S Panangala, F J Hoerr. DOI:
    10.3382/ps.0641678

  * Effects of Purified Aflatoxin on Broiler Chickens.
    J.J.GIAMBRONEU.L.DIENERN.D.DAVISV.S.PANANGALAF.J.HOERR.
    https://doi.org/10.3382/ps.0640852

Author Response

Point 1: Line 17: I suggest changing “scrutinized” and use “examined” instead.

Response 1: Thanks for your nice comments. We have changed “scrutinized” to “examined” (please refer to line 16).

Point 2: These lines need a correction “histological changes company with”, what does it mean?

Response 2: Thanks for your nice comments. We have modified the incorrect description of “histological changes company with” to “Histopathology examinations showed that at day 28, ….” (please refer to line 17). This can be better reflected in the histopathological changes at the same time as the changes in the number and karyoplasmic area ratio in plasma cells.

Point 3: Line 37: Fix this sentence “Atatoxin B1 (AFB1)”.

Response 3: Thanks for your nice comments. We have changed “Atatoxin B1 (AFB1)” to “Aflatoxin B1(AFB1)” (please refer to line 36).

Point 4: Line 41: Delete the first period in “efects. [5,6].”

Response 4: Thanks for your nice comments. We have deleted the first period in “effects [5, 6]” (please refer to line 39).

Point 5: Line 72: Use lower case in “Karyoplasmic area”.

Response 5: Thanks for your nice comments. We have modified the “karyoplasmic area” to be written in lower case throughout the full text (please refer to line 75).

Point 6: Line 81: Use “powder” instead of “pulver” in “Curcumin pulver (2.5%) were….”.

Response 6: Thanks for your nice comments. The mistake has been corrected and we have changed the “pulver” to “powder” (please refer to line 82).

Point 7: Line 121: Please describe brand name of “neutral balsam”.

Response 7: Thanks for your nice comments. The brand name of “neutral balsam” was added into the text (please refer to line 125).

Point 8: Line 98-100: Please add the equipment brand name or model used to mix the feed.

Response 8: Thanks for your nice comments. We have added the model “vertical mixer” used to mix the feed (please refer to line 107).

Point 9: Line 108: Change “by” for “with” in “washed by phosphate buffer”.

Response 9: Thanks for your nice comments. We have changed “by” for “with” in “washed by phosphate buffer” (please refer to line 111).

Point 10: Line 117: Add the exact time in “for more than 24 h at 4℃”. The statement is vague.

Response 10: Thanks for your nice comments. We have added the exact time in line 124 as “for 24 h at 4℃” (please refer to line 120).

Point 11: Line 128: “under the oil lens of Nikon microscope” Please rephrase, perhaps to “…organ was examined with a Nikon (model) microscope under 100X magnification using immersion oil”.

Response 11: Thanks for your nice comments. This sentence was rephrased into “The number of plasma cells in each lymphoid organ was examined with a Nikon microscope under 100X magnification using immersion oil” (please refer to line 131).

Point 12: Line 133: Must be “Sigma-Aldrich, USA”.

Response 12: Thanks for your nice comments. We have revised the brand of TRIZOL “Sigma, USA” to “Sigma-Aldrich, USA” (please refer to line 137).

Point 13: Line 158: Use different wording in “depressed spirit”, maybe lethargy or depressed behavior.

Response 13: Thanks for your nice comments. We have changed the adjective word “depressed spirit” into “depressed behavior” (please refer to line 165).

Point 14: Line 163: Must be “significant reductions” not “were significant decreased”.

Response 14: Thanks for your nice comments. We have changed “significant reductions” into “…were significantly decreased” (please refer to line 171).

Point 15: Line 186: Must be “was not self-recovered in AFB1 group”.

Response 15: Thanks for your nice comments. We have changed “was not self-recover in AFB1 group” into “was not self-recovered in AFB1 group” (please refer to line 195). The manuscript is thoroughly modified accordingly.

Point 16: Line 207: Eliminate “th” to read “35 days”. Check appropriate use of “th” in all parts of the text.

Response 16: Thanks for your nice comments. We have eliminated “th” to read “35 days” (please refer to line 199). And the manuscript is thoroughly checked and modified accordingly.

Point 17: Line 228: Use lower case T in “group, The present”.

Response 17: Thanks for your nice comments. We have modified the word “T” in “The present” in lower (please refer to line 237).

Point 18: Line 294: “Consist with our results,” needs rephrasing.

Response 18: Thanks for your nice comments. We have rephrased this sentence (please refer to line 309).

Point 19: Line 322: Add what “Se” stands for, as it is the first time is introduced in the text.

Response 19: Thanks for your nice comments. “Se” stands for the “trace element selenium”. We have added the full name of “Se” in the sentence (please refer to line 343).

Point 20: In the discussion there needs to be a brief explanation on why using 5 mg AF/kg feed is relevant. Please add references to support the dosage used and how this can be comparable or not to realistic conditions. Is this frequently observed in some countries? Please add more details.

Response 20: Thank you for your precious comments on this paper. Importantly, in our pre-experiments, we administered 1, 3 and 5 mg AFB1 per kg feed. On the basis of our preliminary data, we choose 5 mg/kg AFB1 dose. However, the dose (5mg/kg) of AFB1 also present in previous literature. Fernandez et al., administered three dietary treatments (0, 2.5 and 5 mg aflatoxins per kg feed) to broiler and laying hens for 32 days. (Fernandez A, Verde MT, Gascon M, Ramos J, Gomez J, Luco DF, Chavez G. Variations of clinical biochemical parameters of laying hens and broiler chickens fed aflatoxin-containing feed. Avian Pathol. 1994 Mar;23(1):37-47). The corresponding discussion has been added to the article (please refer to line 314). Regarding to aflatoxin contamination, it often occurs in developing countries and moist area, the concentration of AFB1 contaminated feed is higher, and poisoning events caused by contaminated feed are relatively frequent, such as India, Ethiopia, Pakistan, etc.

Point 21: Figure 1 needs improvement. The asterisks need to be larger and there is a hash (#) instead of different letters denoting significant differences. Same comment for figures 2 to 5.

Response 21: Thanks for your suggestion. We have improved the Figure 1-5, and modified the asterisks in Figures. The comments for all the Figures have been revised carefully (please refer to Figure 1-5).

Point 22: Please add 1 or 2 relevant histopathological images of lesions.

Response 22: Thanks for your nice comments. We have added the histopathological images of lesions as supplementary Figure 1-6. Relevant pathological lesions have been indicated using colored arrows at the appropriate locations in supplementary Figure 1-6. Please refer to the supplementary materials.

Point 23: Relevant pioneering papers need to be included, for instance:

* Effects of aflatoxin on young turkeys and broiler chickens. J J

Giambrone, U L Diener, N D Davis, V S Panangala, F J Hoerr. DOI:

10.3382/ps.0641678

* Effects of Purified Aflatoxin on Broiler Chickens.

J.J.GIAMBRONEU.L.DIENERN.D.DAVISV.S.PANANGALAF.J.HOERR.

doi.org/10.3382/ps.0640852

Response 23: Thanks for your nice comments. Relevant pioneering papers have been added into the discussion part (please refer to line 312).

Reviewer 3 Report

Abstract: This section has been written very weak. In addition to ambiguous words and sentences, the authors should clearly write their important findings. At present, the abstract section does not represent the essence of this study. In addition, the conclusion is also very ambiguous.

Introduction: In this section, the authors should tell what is missing in the previous studies and what they are intending to do in the present study. This section needs coherence which is missing and major drawback.

Materials and methods

The number of birds and the subsequent replicates are few and it is not enough to make any conclusion.

What is the reason of curcumin 450?

The methodology of AFB1 in the feed is very crude and does not carry any reference. Was it sprayed on the ready to eat feed or stored feed?

What is the basis of resampling at 35 days?

Why histopathological pictures were not presented?

Results

In this section, the figures have been presented so poorly that it is difficult to find what is there in the text.

Is it not better to show them in a table?

Discussion: the discussion is more a review of literature than a plausible explanation. The authors should focus on the real explanation of the results and not try to increase the length of the discussion with a review of the literature.

Author Response

Point 1: Abstract: This section has been written very weak. In addition to ambiguous words and sentences, the authors should clearly write their important findings. At present, the abstract section does not represent the essence of this study. In addition, the conclusion is also very ambiguous.

Response 1: Thanks for your precious comments on this paper. In the revised paper, we have remodified the abstract part to eliminate the ambiguous words and sentences. The aim, hypothesis and significance of research were emphasized, and the important findings were clearly reflected, which can better represent the value of the work. Also, the conclusion was shortened and clearly stated.

Point 2: Introduction: In this section, the authors should tell what is missing in the previous studies and what they are intending to do in the present study. This section needs coherence which is missing and major drawback.

Response 2: Thanks for your precious comments on this paper. We carefully reviewed the introduction and provided supplementary detail to the missing parts. The shortcomings of the previous study were explained, and the purpose and what we are intending to do in this study were clarified. The introduction is thoroughly revised, corrected and minor mistakes are corrected accordingly.

Point 3: Materials and methods: The number of birds and the subsequent replicates are few and it is not enough to make any conclusion.

Response 3: Thanks for your precious comments on our paper. In our pre-experiments, 20 broilers were randomly divided into each group and all aspects of data was stable. It indicates that the model of curcumin alleviating AFB1-induced toxicity in broilers was successfully established. In this paper, 96 broilers were randomly divided into 6 groups (n = 16), and the relative experiments were repeated three times after sampling, the stability and reliability of data can be guaranteed.

Point 4: What is the reason of curcumin 450?

Response 4: Thank you for your precious comments on this paper. Importantly, in our pre-experiments, we administered 1, 3 and 5 mg AFB1 per kg feed. On the basis of our preliminary data, we choose 5 mg/kg AFB1 dose. Then, we administered 150 mg/kg, 300 mg/kg and 450 mg/kg curcumin to alleviate AFB1-induced varieties of organs (liver, kidney, spleen, etc.) injury, the results showed that curcumin can protect broilers against AFB1-induced toxicity in a dose-dependent manner and 450 mg/kg curcumin shows the best protective effects. There also other researches chosen more than 450 mg/kg curcumin to prevent AFB1-induced acute toxicity (Jin S, Yang H, Jiao Y, Pang Q, Wang Y, Wang M, Shan A, Feng X. Dietary Curcumin Alleviated Acute Ileum Damage of Ducks (Anas platyrhynchos) Induced by AFB1 through Regulating Nrf2-ARE and NF-κB Signaling Pathways. Foods. 2021 Jun 14;10(6):1370. doi: 10.3390/foods10061370.; Jin S, Yang H, Wang Y, Pang Q, Jiao Y, Shan A, Feng X. Dietary Curcumin Alleviated Aflatoxin B1-Induced Acute Liver Damage in Ducks by Regulating NLRP3-Caspase-1 Signaling Pathways. Foods. 2021 Dec 13;10(12):3086. doi: 10.3390/foods10123086.). Moreover, the dose chosen and the application of 450 mg/kg curcumin was also published in other papers by our team (Cheng P, Ishfaq M, Yu H, et al. Curcumin ameliorates duodenal toxicity of AFB1 in chicken through inducing P-glycoprotein and downregulating cytochrome P450 enzymes. Poult Sci. 2020;99(12):7035-7045. doi:10.1016/j.psj.2020.09.055; Wang H, Muhammad I, Li W, Sun X, Cheng P, Zhang X. Sensitivity of Arbor Acres broilers and chemoprevention of aflatoxin B1-induced liver injury by curcumin, a natural potent inducer of phase-II enzymes and Nrf2. Environ Toxicol Pharmacol. 2018;59:94-104. doi:10.1016/j.etap.2018.03.003).

Point 5: The methodology of AFB1 in the feed is very crude and does not carry any reference. Was it sprayed on the ready to eat feed or stored feed?

Response 5: Thanks for your nice comments. We have carefully checked and modified the methods about the feed production. Briefly, AFB1 was dissolved in 30 ml methanol, then this prepared solution was sprayed evenly on the basal diet and mixed to obtain the 5 mg/kg AFB1-contaminated diet. The equivalent methanol was sprayed evenly on the normal feed to obtain the AFB1-contaminated diet. The treatment concentration of curcumin was calculated and added uniformly into the diet and mixed evenly. For instance, 18.0 g curcumin (2.5%) powder was weighed and added into the basal diet, then the diet was mixed thoroughly to obtain the 450 mg/kg curcumin-contaminated diet. All the diets were evaporated at 37 °C and mixed with a vertical mixer. And the reference was added in the 2.2. Chickens and Treatment (please refer to line 102).

Point 6: What is the basis of resampling at 35 days?

Response 6: Thanks for your nice precious comments. Each group kept on a corn-based commercial diet (aflatoxin binder free diet) for 28 and 35 days. After 3 days of acclimatization, six groups’ broilers were provided corresponding diets for the next 28 days, while from day 28 to day 35, normal feed (only basal corn diet without AFB1 and curcumin) were provided for all six groups. Resampling was performed at 35 days to explore whether self-recovery or post-toxic effects could occur after AFB1 administration was withdrawn, and whether the curcumin intervention was sustained for 7 days after the withdrawal of curcumin.

Point 7: Why histopathological pictures were not presented?

Response 7: Thanks for your nice comments. We have submitted the histopathological pictures as supplementary materials. The histopathological changes in pictures were shown by colored arrows and annotated in comments. Please refer to supplementary materials (pictures 1-6).

Point 8: Results: In this section, the figures have been presented so poorly that it is difficult to find what is there in the text. Is it not better to show them in a table?

Response 8: Thanks for your nice comments. We have carefully checked the data and revised the figures as another form to better reflect the results. However, we believe that pictures can more intuitively reflect the data than tables.

Point 9: Discussion: the discussion is more a review of literature than a plausible explanation. The authors should focus on the real explanation of the results and not try to increase the length of the discussion with a review of the literature.

Response 9: Thank you very much for your nice comments on our article, which were very useful for improving this draft. We thank you for spending your precious time on this paper. We have carefully revised the discussion part. For the cited literature, it is not only to review what other researches have done, but also to explain our work by combining their findings and theory. Further, by comparing their results with ours, the effects of different species, different AFB1 doses or different detoxification drugs on immune organs could be revealed, and finally showed the protective effect of curcumin on AFB1 immunotoxicity in this test. The added and modified part of discussion have been indicated in red color.

Round 2

Reviewer 1 Report

In Point 1 I have asked for a valid motive to not include the control groups with 150 and 300 mg/kg of Curcumin. I understand that these lower doses are safe, however, the authors could improve their analysis if the include  the control groups with 150 and 300 mg/kg.

Figures were improved but I am not sure about the significant differences between AFB1 group and control group, and AFB1 and the other groups. All the figures show significant differences between AFB1 and the control (*) and between AFB1 and the other groups (#). These differences are not evident in a lot of graphs. For example: In figure 1C, values of AFB1 present significant differences to curcumin I? The same for: 2A, 3A, 3B, 4B, 5B. That is the most worrying point of the work.